# Couple-centered HIV prevention and care: Endorsement, practice and uncertainty among us healthcare providers in western-central upstate New York

Natalie M. Leblanc [1,2]*, Sadandaula R. Muheriwa-Matemba [3], Noelle St. Vil [4], Danielle Alcena-Stiner [1,2], Keosha T. Bond [2,5], Alexander Glazier [6], Luis Rosario-McCabe [1], Faith Lambert [1], Martez Smith [7]

1 School of Nursing, University of Rochester, Rochester, New York, United States of America, 2 Research Education Institute for Diverse Scholars (REIDS), Center for Interdisciplinary Research on AIDS, School of Public Health, Yale University, New Haven, Connecticut, United States of America, 3 Department of Human Development Nursing Science, University of Illinois, Chicago, Chicago, Illinois, United States of America, 4 School of Social Work, University of Buffalo, Buffalo, New York, United States of America, 5 School of Medicine, Community Health and Social Medicine, City University of New York, New York, New York, United States of America, 6 Department of Public Health Sciences, University of Rochester Medical Center, Rochester, New York, United States of America, 7 HIV Center for Clinical and Behavioral Studies, Columbia University, New York, New York, United States of America

* natalie_leblanc@urmc.rochester.edu

## Abstract

Given the influence of provider perspective and practice in the uptake of HIV/STI prevention and care strategies, this qualitative descriptive design study sought to illuminate perspectives of couples HIV testing and counseling (CHTC) and describe couple/partner-based practices among health providers in New York State. We utilized a purposive sampling strategy to recruit health providers (N = 27). Semi-structured in-depth interviews were conducted from Sept. 2019 to Feb. 2021. Four themes emerged: perspectives on engaging partners and couples-centered sexual health promotion; providers' experiences with patients and partners in HIV prevention and care; provider endorsement of CHTC; and perceived CHTC implementation determinants. CHTC endorsement was prominently due to the perception of CHTC as a facilitator to enhance patient-provider engagement in HIV/STI treatment and care, especially in the communication and dissemination of information among partners. Providers reported that health literacy needs regarding HIV/STI testing and diagnosis, but primarily STIs treatment regimens warranted a joint approach. CHTC endorsement entailed the strategy's perceived ability to enhance sexual health literacy among patients and patient's partners. Lastly, CHTC endorsements entailed provider beliefs that it ensured knowledge equity and joint literacy in the communication of health information among health consumers. Determinants of CHTC implementation were factors that providers perceived to have a bearing on the facilitation or posed as barriers to jointly engage partners in HIV/STI prevention and care and was subsequently a source of provider uncertainty. These determinants ranged from provider-level factors to organizational capacity issues that could impact CHTC implementation. Recommendations for CHTC are discussed.

**Data availability statement:** All relevant data are within the paper and its Supporting Information files.

**Funding:** This pilot study was supported by NIH/NIMH R25MH087217-09. The funders had no role in study design, data collection and analysis, decision to publish, or preparation of the manuscript.

**Competing interests:** The authors have declared that no competing interests exist.

## Introduction

Efforts to end the HIV epidemic (EtHE), must address persistent contributors to ongoing HIV transmission at different levels of influence. Sex-based HIV transmission (penetrative vaginal and anal) dominates the mode of HIV acquisition in the U.S., with 70% of transmission occurring between male same-sex partners, and 22% were among hetero-sex partners [1]. The Centers for Disease Control and Prevention show that regardless of sex, sexual orientation or gender expression, a quarter of people living with HIV (PLWH) were not diagnosed, another quarter diagnosed did not receive care, and a third were not virally suppressed [2]. Furthermore, regardless of sexual orientation, it is estimated that 40–60% of HIV transmission occurs among intimate partners known to be already living with HIV infection. Demonstrating that interpersonal factors (such as behaviors within couples) may pose as the most persistent and more proximal determinant of ongoing HIV infection.

Interpersonal relationships like those between health providers and their patients, in tandem with provider characteristics, perspectives and certainty concerning bio-medical interventions, are determinants of health consumer awareness, willingness, and access to HIV interventions. For example, one study [3] found that African-American female adolescents perceived health providers as being judgmental and uncomfortable when discussing STIs susceptibility, which resulted in these adolescent girls hesitation to disclose health information in the clinical setting. Leblanc et al. (2019) reported provider perspectives on implementing couples-centered HIV testing and counseling CHTC. They found that providers hesitation was in part due to their colleagues' discomfort in the provision of HIV screening and care, which would subsequently impede any efforts to adopt recommended guidelines to engage people and their partners in joint screening [4]. Another study [5] demonstrated that providers in training held racist beliefs that resulted in presumptive withholding of PrEP from Black MSM. These findings demonstrate the role of provider comfort, perspectives, biases, and practice on the potential and actual uptake of HIV and related biomedical interventions. Healthcare providers make clinical judgments about their patients access to and eligibility for the myriad of HIV behavioral and biomedical interventions.

A second dimension of interpersonal relationships that impacts HIV prevention and care, is that between intimate partners. Couples, and other self-defined intimate partners need to be considered for intervention and biomedical support. This is due to ~ 50% of new HIV infections in the U.S. are partner-based, and that among those newly diagnosed with HIV were with partners known to be living with HIV infection. Additionally, increasing rates of STIs, require a more robust approach beyond existing partner therapy modalities [6,7]. Further, the motivation for being in a relationship and to engage in behaviors as intimate partners are known determinants of the uptake of HIV prevention interventions [8].

Couple's HIV testing and counseling (CHTC) began as an opportunity to facilitate disclosure following HIV screening, and largely implemented in antenatal services outside of the U.S [13–18]. CHTC has evolved as a strategy to address the interpersonal attributes of HIV susceptibility, persistent transmission, and suboptimal engagement in HIV care. This strategy has gathered considerable empirical support for its effectiveness and efficaciousness in reducing the number of sexual partners in tandem with condomless sexual acts. CHTC has demonstrated an increase in the ease of partner disclosure among those living with HIV, and sustaining linkages to medical care for those who are diagnosed with HIV [9–14]. It is a mechanism that has been shown to optimize biomedical advances like rART, pre- and post-exposure prophylaxis (PrEP, PEP), and various HIV screening modalities (i.e., home-based HIV screening) [15]. Furthermore, it has been successful among male same-sex couples in reducing substance use and sex-based HIV susceptibility in U.S. settings, and there is acceptability (albeit no real sustained effort) for such

an approach among U.S. heterosexual couples [16–18]. Despite 30+ years of evidence, health provider and consumers globally endorsing this approach, and the existence of U.S. national implementation guidelines, couple-centered approaches to HIV prevention and care remain underutilized and implementation experiences are not well documented in the U.S. [13,19].

Given the critical role of health providers in the uptake of HV prevention and care strategies, the World Health Organization recommends that ascertainment of healthcare provider perspective on CHTC and practice is imperative to inform intervention development and implementation. The literature on providers' perspectives of couples-based HIV testing and counseling (CHTC) in the U.S. is scant, but previous work has showcased provider perspectives and willingness to pursue CHTC as a modality to address ongoing community level transmission [11]. Uptake of CHTC and other partner-based services like expedited partner therapy in the U.S. are challenged. This is in part due to a Western-based approach to medicine, a for-profit driven healthcare system that engages individuals without full consideration for their relationships, and providers not adequately trained or have restrictions imposed on their scope of practice [20]. Additionally, guidance on implementation in the U.S. may be inadequate due to the diversity in sexual activity and subsequent disparity among health consumers who would benefit from these services. Also, health settings may not be well equipped to implement comprehensive and HIV status neutral services, and a U.S. health service reimbursement structure may not adequately support interpersonal factors associated with health promotion and disease prevention. Therefore, this study builds on seminal work in the Southeast U.S. [4] and seeks to characterize provider perspective on CHTC and experience with engaging partners in WCNY State. Such knowledge is necessary to address a critical gap in evidence-based interventions aimed to enhance provider interactions with couples and patients who are partnered seeking HIV prevention and care services to reduce HIV infections. This knowledge is essential for intervention development, and training healthcare providers to identify and overcome sources of discomfort, and biases in their interactions with people and subsequently couples in need of HIV prevention services and care. To that end, the purpose of this paper is to elevate an HIV screening modality that is underutilized in the U.S. and describe CHTC implementation considerations in health settings. Given the influence of provider perspective and practice in the uptake of HIV/STI prevention and care strategies, this report aims to illuminate perspectives of CHTC, and describe couple/partner-based practices among health providers whose engage in sexual and reproductive health service provision in WCNY State.

## Materials and methods

We approached this inquiry with a qualitative descriptive design. This allowed for a direct description of providers perspectives and description of their subjective experiences [21]. Additionally, it's increasingly an approach used in subsequent intervention development [22]. Although there is evidence that CHTC is an effective strategy, we approached data collection to include inquiry about recommendations for implementation in a setting that is different from a large urban center like New York City.

This study is situated in the Western/Central parts of New York (WCNY) State. WCNY is comprised of both a significant rural population and small cities or urban centers. It contains 4 subregions, 17 counties, and approximately 7 small cities. This part of New York State has closer proximity to Southeast Canada, than to New York City. These small urban centers/cities are recovering from deindustrialization with the influx of emerging industries, has one of the highest levels of child poverty in the country, and experiences substantial increases in HIV/STI rates [23]. The ethno-racial population in this region is

predominately White-identified, and Black populations make up about 1/3 in the small urban centers [24]. In New York State (excluding New York City), the context of HIV transmission in WCNY State is seen primarily among an older adolescent/young adult population, in the midst of a significant increase in other STIs including congenital syphilis (even before the COVID pandemic commenced) [25]. The area is also experiencing an increase in mortality related to substance use, and a substantial increase in community-level violence including hate crimes, state-sanctioned and civilian murder [26]. Health policy changes in HIV/STI care, like updated local guidelines for expedited partner therapy (EPT), have not been optimized and more contextual approaches are warranted in efforts toward ending the HIV epidemic [27].

A qualitative descriptive design allowed for flexibility in decision making as the research process progressed and a representative of a range of perspectives. In alignment with our approach, we utilized purposive sampling to recruit health providers. Employing maximum variation techniques [22], we recruited providers during grand rounds or meetings with leadership at local health facilities, community forums, via online forums, professional networks, and snowball sampling. Facility based recruitment sought settings in which sexual and reproductive health services are either the main service provided or one of the main services offered. Potential subjects were invited to contact research team members via email and complete a study information form that included additional details about the study, solicited provider demographics and requested them to indicate their willingness to participate in an in-depth interview. This served as their consent to participate in the study. Once subjects indicated their interest and completed the demographic form, indicated their willingness to participate in the study via the information sheet, an interview time was scheduled via zoom.

In-depth interviews were facilitated by a semi-structured topic guide and included open-ended questions about sexual health promotion, HIV/STI prevention and care, and implementation (see S1 File. Interview guide and label). All interviews were conducted virtually by the lead principal investigator and two research assistants at a time most convenient to the healthcare provider, to accommodate healthcare provider schedules and the onset of COVID-19 restrictions. Interviews were audiotaped to capture participants insights and allowed a focus on active listening, probing, and maintaining eye contact via zoom, to foster rapport and open in-depth conversation. The data collection and analysis were concurrent processes [28] and data was managed in MAXQDA. Thematic analysis commenced with a review of the interview guide and topics; a review of a subset of transcripts; and ongoing peer debriefings to ensure that the meaning, interpretation, and analytic approach were uniform among the primary analysis team [29]. Analysis involved reading and re-reading transcripts to develop a coding structure that reflected both deductive categorizing of topics identified a priori and inductive emergent thematic patterns in the data, resulting in 4 thematic categories.

Following all interviews, settings who were approached for recruitment and who agreed, received an oral report back of findings.

## Results

Providers (N = 27) were recruited (September 2019 – February 2021) from Western/Central New York State representing an array of medical providers. Providers were mainly female (N = 23), aged 24 to 60 years old, identified primarily as Christian, and ranged in practice from 1 to 25 years are other demographics are detailed in Table 1. Four thematic categories pertaining to healthcare provider perspectives on CHTC and experiences with a partner-centered approach: Perspectives on engaging partners and couples-centered sexual health promotion; Providers' experiences with patients and partners in HIV prevention and care; Provider

**Table 1. Provider demographics.**

| Demographic | | N |
|---|---|---|
| Age | <30 years old | 8 |
| | 31–49 years old | 10 |
| | >50 year olds | 9 |
| Sex | Female | 23 |
| | Male | 4 |
| Ethno-racial identity | Hispanic | 1 |
| | Black | 1 |
| | Asian | 1 |
| | American Indian/Alaskan | 1 |
| | White | 23 |
| Provider type | Advance Registered Nurse Practitioner (ARNP) | 12 |
| | Registered Nurse (RN) | 3 |
| | Physician/Medical Doctor (MD) | 7 |
| | Physician Assistant (PA) | 4 |
| | Medical Assistant (MA) | 1 |
| Provider specialties | OB-GYN/Midwife | 2 |
| | Family Medicine/Infectious Disease | 7 |
| | Pediatric/Adolescent health | 1 |
| | Women's Health | 6 |
| | Generalist/Sexual and Reproduction | 10 |
| | Psychiatry/Mental Health | 1 |
| Provider Setting | Hospital-affiliated clinic | 9 |
| | Federally qualified health center (FQHC) | 3 |
| | STI clinic | 4 |
| | Community health clinic/private practice/non-profit | 10 |

endorsement of couple-centered HIV testing and counseling (CHTC); and Perceived CHTC implementation determinants.

For the exception of two medical doctors with over 20 years' experience each, no participant had heard of CHTC or the Center for Disease Control and Prevention (CDC) program, *Testing Together* or any other iteration of this strategy. Before discussing their perspectives on CHTC, providers were asked about their general perspectives on engaging couples or partners in care and sexual health promotion and HIV specifically.

Overall, perspectives contained a range of reflections about this approach that were seemingly influenced by actual experiences with considering a patient's partner, engaging couples jointly or reflecting on a hypothetical situation in which a partner-based or couple-centered approach would be warranted.

## Perspectives on engaging partners and couples-centered sexual health promotion

Providers were asked about their perspectives on engaging patients and their partners (couples) in sexual health services. Overall providers endorsed engaging couples and utilizing a joint approach based on their own experiences with patients. They endorsed engaging patients and their partners due to what they believed to be the ability to enhance provider-patient communication, a time-efficient approach for communicating key messaging and enhance care.

> "I just love the idea of bringing people together, and instead of saying it eight times, saying it fewer times, in this case, instead of saying it twice to two separate people, saying it once to them both and then they both get to hear each other's questions and answers…"

> ARNP, 10 yrs., Women's Health

Providers also perceived partner-based approaches as ensuring that both the patient and the patient's partner understand health information, especially in the context where some information regarding treatment and transmission for certain STIs can be misunderstood. This was perceived as being of added significance in the patient-provider interaction and communication, especially when there are health implications for the partner. Further, providers perceived that jointly engaging partners would not only ensure health information is accurately disseminated to the partner of their patients and presumably jointly understood but was perceived to discuss complicated health issues. Providers noted that such communication about difficult issues may sometimes warrant both partners to be engaged in a health visit to ensure that a particular health intervention is accurately considered and to facilitate uptake.

> "I can see the benefits of seeing couples together, especially for something like PrEP…I can see how it can be helpful, and maybe even enhance compliance with taking medicine some things like that if partners are involved in their care as well."

> "I try really hard to invite partners into the room for any kind of visit that people feel comfortable having…I think it's important to make space for people to have private conversations and…to have support people there with them to help them remember recommendations and talk about things in a way that helps it cement in their head…it's a really frequent thing that I'm saying I wish that I have a partner here in the room with them to navigate this conversation…"

> MD, 9 yrs., OB/GYN

## Providers' experiences with patients and partners in HIV prevention and care

Several providers reported experiences whereby they either engaged individuals in HIV prevention and care which included a focused conversation regarding partners (partner-based); or a more explicit engagement with both partners in the same space and time (couple-centered). In a partner-based approach, providers reported on conversations and practice that revolved around patient education about HIV/STI treatment, transmission, and partner implications. Providers described a more couple-centered approach focused on actively engaging both sex partners or couples in joint conversations or their practice to address the presence of or susceptibility to HIV or STI acquisition.

Partner-based engagement consisted of provider communication with patients about partners. Providers described that some health assessment information had to be obtained through one-on-one conversations since there were issues that would possibly not be addressed with both partners. These conversation with patients however considers the role of partners, like intimate partner violence, asking about non-monogamous relationships or the involvement of people not included in the main partnership.

> "…it's very important to have that one on one interaction without a partner there, for doing things like you know asking about other exposures, outside of that relationship and safety in sexual relationships and intimate partner violence, things like that, that

you can't ask in front someone else…and I am not infrequently in the position of having conversations with couples together, where either I or one of my [practice] partners is the doctor for both members…one partner of the couple is my patient, and the other one doesn't have a physician"

<div align="right">MD, 9 yrs., Family Medicine</div>

Providers reported experiences with patients regarding recurrent diagnoses of sexually transmitted infections or conversations to facilitate partner disclosure to partners and partner treatment. Providers were also responsive to an index patients request for assistance in communicating with their partner about their STI.

"…herpes and then also with genital warts, these are the ones where it often comes where I will have a conversation… sometimes I will invite the partner to come in the exam room… I've done that also on the phone… that will be for patients who…don't know how to talk to their partner or their partner has a ton of questions… I think it just helps the patient…they don't remember everything I am saying, so it helps them to answer questions, or understand the ongoing ramifications…"

<div align="right">ARNP, 10 yrs., Women's Health</div>

In the context of a new STI diagnosis, providers engaged patients in conversation to facilitate engagement in care, treatment, disclosure, and partner screening. A partner-based approach also included education to help patients understand the transmission and treatment of certain STIs over others (HIV and herpes versus Chlamydia and Gonorrhea) and the implications for partner-based transmission.

"Honestly, the worst diagnosis to tell anyone about is herpes…people cry, people think their lives are over……herpes has like the worst stigma of any STI and I also think it's not well understood …. I think people have this idea…they won't be able to sex with anyone again…it may be just misunderstandings about herpes…and just the idea of like, 'how am I gonna tell my partner?' 'And when did I get this?'

<div align="right">ARNP, 10 yrs., Women's Health</div>

At other times, partner-based engagement included providers whose communication was with patients whose greatest concern was maintaining the integrity of their intimate relationship. Relatedly, providers also reported engaging with patients who utilized HIV screening as an opportunity to facilitate disclosure of their known HIV serostatus to a partner. This approach may include facilitating partner screening for HIV, in which the interaction between the patient and provider may morph into a more couple-centered approach whereby a patient and partner are seen jointly.

"…we have somebody who is truly newly diagnosed [with HIV] who is devastated and has the fears of rejection and "How am I going to tell my partner?" and all of this. Then you have the person who is a known positive, who's been afraid to tell their partner. And we work with that person to try to find a way and sometimes those people will come in and have a test, even though they're a known positive, and claim it to be a new diagnosis so that they can inform their partner that they have HIV."

<div align="right">RN, 6 yrs., Sexual/reproductive Health</div>

Couples engagement was also perceived to be a way to get a sense of a patient's support system and better contextualize their interpersonal lives. This insight influences the provider-patient relationship and subsequent care for the patient.

> "…people are already going to each other's visits and it's great… it augments health care if there's 2 people…I've almost always found it helpful when a partner is there to give us a little bit more information…like last week this postpartum lady had depression … her husband knew a lot more detail ….it gave me an idea of what type of support she had at home… I was talking to both of them equally about what they can do to help each other and especially during this time that was really challenging."

> <div align="right">ARNP-Midwife, 8 yrs., Sexual/reproductive Health</div>

## Provider endorsement of couple-centered HIV testing and counseling (CHTC)

**Endorsement of CHTC.** Overall, providers endorsed CHTC as a viable strategy and described their willingness to implement CHTC. However, some providers also expressed uncertainty alongside their endorsement.

*CHTC endorsement*: Though most providers had no previous knowledge or had even heard of CHTC, many endorsed the strategy and generally perceived its implementation as feasible in their health settings. Provider endorsement is characterized as having support for a couple-centered approach broadly and CHTC specifically while citing fundamental concerns and possessing limited reservations about its implementation. The most positive endorsements of CHTC cited that it ensured knowledge equity and joint literacy in the communication of health information among health consumers.

Endorsement was primarily informed by experiences with patients.

> "…it [CHTC] would be really helpful, because...…it's a lot to take in a visit… we're treating them here today and they need to abstain from intercourse for a week. And then they go and for whatever reason they only absorb that they need to abstain from intercourse for 3 days…even though they were treated, now their partner has it or they're re-infected…it would be nice if they were involved in the same conversation… sometimes it's nice to hear it together."

> <div align="right">ARNP, 6 yrs., Women's Health</div>

Endorsement was also informed in some cases, as reflected below, provider frustration with persistent HIV and other STI transmission.

> "it's critical…what we really need to do is facilitate people sharing their results… I'm so tired of treating people for the same thing over and over again…it's like the message is not getting communicated to both of them …I've heard patients like 'oh he told me it's a UTI' and it'll be gonorrhea …this is not sustainable, we need to get you treated or you need to stay away from this person… I think having people to coach HIV testing together, would actually facilitate that sharing of results a little bit and maybe even strengthen their interpersonal relationships so that they can talk about those things and other things, so I would love that (CHTC) that sounds great."

> <div align="right">ARNP-Midwife, 9 yrs.</div>

They further reported that CHTC would allow partners to advance discussions about their sexual health more broadly. CHTC could also facilitate conversations between partners that otherwise would not happen.

"…. if it's done the right way, approached at the right way, then do it [CHTC]. Because people in relationships, I mean there is some relevance….and I realize there are traps here, but in some settings and in some situations two people who hear the same information at the same time from the same person provides an opportunity to have some discussion and …learn something.

MD, >20 yrs., ID medicine

***Uncertainty with CHTC endorsement***: Some providers generally endorsed CHTC, but expressed some uncertainty based on their experience with patients and their partners, and the perceived nature of partners and relationships. Other sources of uncertainty were based on knowledge of their patient population and questioning whether there would be a demand for CHTC.

"I see a fair number of couples, even the couples that self-identify as couples have a fair amount of distrust of one another, and I don't think that it would be a reason not to offer it, but I do wonder how many people would be willing to take that strategy, knowing that it requires mutual disclosure."

MD, 9 yrs., Family Medicine

Other providers who endorsed CHTC were also attentive to relationship dynamics and other relationship attributes as part of an overall assessment of the couples' health and well-being. Some of the uncertainty in these cases were based on known HIV stigma that may impact relationship integrity and the type of couple who would or should engage in CHTC.

…this could be a great strategy for a good relationship.…if it's a casual relationship … I may have a bias...it makes sense intellectually, it makes perfect sense, but I feel my bias… if I'm in a new relationship, and I find out that this person I'm with has HIV, am I going to want to continue in this relationship?...I want to enter a relationship with someone knowing they are HIV positive, I wouldn't, which is terrible for me to say, because that's not how I counsel patients."

ARNP, >10 yrs., Women's Health

Sources of provider CHTC uncertainty was also demonstrated in considerations for implementation. Providers speculated that some information about the couple's relationships and/or individual partner behaviors might not be revealed in joint sessions or health visits. Therefore, separate sessions would elicit information that would otherwise not be shared due to discomfort or lack of transparency between partners. Providers perceived the omission of certain information would give insight into the relationship dynamics, health needs, or HIV susceptibility within the couple.

"I feel like you could do it [CHTC] as long as you still had a component when you separate them [and] talk to them separately. Because if there are other partners involved or other like sexual behaviors involved that they're not comfortable sharing… they might

be less willing to express that. So, you might get a sort of skewed risk assessment…it all depends on the status of the relationship. But it sounds like a good idea."

MD, 2 yrs., Infectious Disease

**CHTC implementation determinants.** Determinants of CHTC implementation were factors that providers perceived to have a bearing on the facilitation or posed as barriers to jointly engaging partners in HIV screening. These determinants ranged from provider-level factors to organizational capacity issues that could impact CHTC implementation.

*Provider willingness to implement CHTC:* Provider willingness to implement CHTC: Despite not having previous knowledge about CHTC, most providers reported they were willing to implement CHTC. Their willingness appeared motivated by desiring to address the HIV prevention needs of the patient population served and having partner engagement experiences. Willingness was partly based on exposure to and their own experiences of considering partners in their practice or engaging in a partner-based or couple-centered modality. Other providers simply believed that a joint patient and partner consultation allowed an opportunity for discussion of health topics in the presence of a health practitioner. The practitioner represented a third party and was perceived to be a buffer in any conflict between partners outside the clinical setting.

"I'm very willing to do whatever it takes to get people the right information, because sometimes you have- two people get a visit and you tell both of them the same thing and they both can interpret it differently and then they go home together and then they're still blaming each other…whatever it takes to get the right information to them… engaged in whatever care it is that they need, I'm willing to be flexible to do that."

ARNP, 10 yrs., Women's Health

Some providers were willing to implement CHTC because they perceived it to be more efficient in providing patient education. Their willingness appeared grounded in a general desire to provide patients with the best care possible, which involved educating them in a way that was salient for the couple's joint understanding and addressed follow-up to patients in real-time. However, joint HIV testing must be contingent on couples' willingness to engage. Essentially, if the couple was willing, so was the provider.

"I definitely think it's better if you can jointly test them and if they're willing to disclose to each other…that's so much easier than…testing them separately…sometimes I'll be seeing a husband and a wife for...their annual visits, and they want to be seen together… so that would kind of be an extra visit…some people are very open to doing that with each other…I don't have any problems with it."

MD, <5 yrs., Women's Health

Some providers were willing to implement CHTC in their practice and gave insights for doing so. They preferred individual consultations with either partner first, prior to a joint session with both partners to assess the relationship. The prominent reason was to address individual health needs and ascertain a more complete context of the couple's relationship.

"…when two couples are together, they don't say things or they hide things from each other, so I would like to first individually approach it, and then potentially putting them together and see how the dynamic is between those two and see how receptive they are

or how involved they are in the care, but I like, I like the idea of having two of them, both couples together…"

ARNP, >10 yrs., Family Medicine

"…there's lots of patients that do have a partner, an on again off again partner or a long-term partner and it would great to have them be together there. I think how many times a day do you do the same counseling on an individual patient that could've then been done it for two people together. I love the idea to walk into the room and say "Okay so you're both here for testing together, that's so great."

ARNP, 10 yrs., Women's Health

Providers who reported some experience with partners, though endorsed CHTC, appeared to want to maintain some aspects of their current practices to facilitate CHTC implementation. This appeared due to existing protocols that engage partners in a non-threatening way that supports the provider practice.

"…because [of] our protocol we do just kind of start alone at first, just to make them comfortable …there can be a barrier whether they're just really embarrassed or not a hundred percent confident in telling their partner "hey no I want to ask these questions or be counseled by myself" and we find that sometimes they'll be a little more truthful and open in that type of setting…"

MA, <5 yrs., Sexual/Reproductive

Ultimately, provider willingness among participants provided insight into considerations regarding CHTC implementation. As the previous exemplars note, starting a health visit with partners individually was perceived to give a provider better insight into the relationship dynamics and HIV susceptibility. Following individual patient engagement, a joint session would give greater context to the couple's need and ability to receive information that is salient for them.

***Provider capacity to implement CHTC:*** Provider capacity to implement CHTC: Provider perspectives on provider capacity to implement CHTC reflected primarily logistical and couple/partner-level factors that served as either facilitators or challenges to the uptake of CHTC. For example, one provider noted that it was simply more convenient for them to test separately and not engage with both partners simultaneously, due to time constraints and scheduling.

"I probably would have just ordered the HIV antibody and had them just go to the lab and have it drawn…. kind of thing because it's just easier. It's really hard to be able to bring somebody in and sit them for counseling. Just because we as providers don't have a lot of time to do that and there's you know it's hard to just bring somebody in."

PA, 6 yrs., Generalist

Other providers demonstrated capacity by referencing previous experiences with couples in general, not just regarding sexual health or HIV counseling. They also reported being compelled to engage partners due to the couple's health situation, their experience with patient engagement and information retentions, or at the request of one or both partners.

"…I actually tried to work with couples here…if there was a situation where we had a patient here that had a partner, they might be on-site or maybe not, I'd invite them to invite the partner in because there are a lot of things that they felt like they couldn't comfortably discuss with the partner or they felt the partner would get angry, and I felt like I could mediate that and work with them together…"

RN, 2 yrs., Sexual/Reproductive

For some providers, capacity was not concerned with or questioned whether they could implement CHTC, but with contemplation of what the strategy could look like in their setting and considerations that would need to be made in their personal space.

"…if I were going to implement something like that…how do I protect [the patient]… how do I make sure I'm testing them each for the right tests…and protecting their confidentiality or like for female bodied patients…I'm often promoting "Do you want any birth control, is there anything else you need? Do you need a pap?"… those are some of things I would be thinking about…Do we separate patients to chat about that?"

ARNP, 10 yrs., Women's Health

"…physicians are bombarded with a lot of information and it's hard to remember in the moment, all of the tools…the biggest challenge is having it [CHTC] be as a resource top of mind for clinicians, particularly clinicians who aren't seeing people with HIV every single day… It's just always so many different competing priorities, …how do you integrate something in such a way that it doesn't seem to other people to be just one more thing?…"

MD, >10 yrs., Family Medicine

Providers felt competent in their ability to facilitate CHTC implementation, even in specialized settings. Providers in these cases reconciled that for CHTC facilitation and implementation to be possible, the health setting they operated within had to make some accommodations change.

"…we need more time, we'd need a space that's just for doing that… I'd have to talk to them about billing for male partners which I have a license to do so that would be within my scope of practice."

ARNP-Midwife, 9 yrs., Women's Health

**Health facility preparedness to implement CHTC**: Health facility preparedness to implement CHTC: The sample of providers were employed in a diverse set of healthcare environments. The majority worked within federally qualified health settings that provided primary health care and served people living with HIV. A lesser number of providers worked within specialized clinics that were couched in university-based or privatized health settings. Some providers worked in up to 2 settings. However, most reported that they were willing to implement CHTC and that their health facility could be willing to facilitate. Several factors were deemed to impact facility preparedness to implement CHTC and ranged from practice logistics to protocol development and community engagement.

"No, so we don't allow patients… in the same exam room…like a couple comes into the clinic together and they're brought to their own exam rooms, their intake is done separately, I see one separately and then go and see the other…obviously I'm not talking about one with the other or anything like that. So they're treated like separate patients completely even though they came in together and know each other is there…I basically pretend like I don't know they're a couple.

<div align="right">ARNP, 6 yrs., Women's Health</div>

Providers who worked exclusively in specified settings whereby a couple-based approach may not be feasible unless the facility greatly expanded its patient focus to service all types of couples (i.e., women's health clinic), in which some providers were still willing to consider:

"…There's some logistics that we would have to set up so that we could do that but I think that it's totally possible………the typical American model is one patient at a time and billing for one patient at a time… so I think we just need to make sure we have the codes for that and that that's going to work because right now we don't have any male patients"

<div align="right">ARNP-Midwife, 9 yrs</div>

*Logistic considerations for CHTC implementation*: Providers reported and seemed to believe that their practice environment could make adjustments to implement CHTC. Agency preparedness to implement CHTC entailed providing capability and comfort to facilitate CHTC, address the need of diverse patient populations, and the provision of basic HIV screening services.

"I think the majority of people have gotten more comfortable with sexual health…I think you need that. I think you need that level of comfort with sexual health. The second thing you need is point-of-care testing. …I think the ideal delivery is in person, and if you want to do it in person, you want to ensure it to the other person."

<div align="right">MD, 9 yrs., Family Medicine</div>

Provider beliefs in the capability of their health facility to implement CHTC was in part contingent on whether there was a need for their facility to adjust existing clinic protocols or logistical shifts to facilitate implementation. Some of the practice adjustments were perceived to be more about patient flow and front-staff preparations.

"…appointment scheduling, I know it sounds like not a big deal, but it always ends up being a bigger deal and then communicating with the folks who make the appointments on the phone…online appointment scheduling as well, so just kind of figuring out what would that look like, would they be able to add that into their system...in terms of the actual education for staff to actually know what to say and do in the room … that the couple's confidentiality is being maintained."

<div align="right">ARNP, 10 yrs., Women's Health</div>

"I would just need front staff to schedule me time to do it.... and give myself a little bit more time so I could answer any of their questions. I probably would send a message to my care manager and have her reach out to the patient and their

partner...I would probably have them come in and be counseled, either by our RN care manager or by myself."

<div align="right">PA, 6 yrs., Generalist</div>

Another element of logistic considerations was related to billing and reimbursement for CHTC services. This consideration was a touchpoint for providers, particularly those who may not have had experience with partners, but also for providers who worked in setting where public federal dollars support the services they provide.

"...if we had a system that's set up for it and a way to bill for it [CHTC] so I think adding in stuff that's really hard in the system in the one that I currently work in without a really good reimbursement or like a reason to do it basically, a branch reimbursement or you know a setting that makes it easy for us to do."

<div align="right">ARNP-Midwife, 9 yrs</div>

Billing was also an important consideration for specialized settings, like women's health, by which a partner who was male at birth could not be seen due to perceived jurisdictional mandates.

"...for our practice it would be billing...this is assuming the partner in this instance is male. So billing is going to be a huge barrier for us trying to work through that...we are not family, it's a women's health practice, we should not be providing sexual health to cisgender men..."

<div align="right">ARNP, 10 yrs., Women's Health</div>

"...when people come into our healthcare setting, they have to register and they have to have their insurance verified, and there is so much bureaucracy...developing a registration process where the couple can register as a couple for a visit, instead of two separate visits... being able to register together would make it much more realistic to then bring that couple in to see a nurse or clinician together..."

<div align="right">MD, <10 yrs., Family Medicine</div>

Provider and staff shortages were reported as another determinant of agency preparedness for CHTC implementation because it currently impacted care for some facilities. One provider spoke to their willingness to provide CHTC but proposed that a team-based effort and a patient engagement script for providers may best prepare a facility for CHTC implementation.

"...I can do it, but I'd still like some help...I don't do the rapid test, I do the preliminary piece and the dispensing advice … so the tester person also needs to be prepared...I'm okay with parts of it [CHTC] being scripted or being scripted for some people."

<div align="right">MD, 20 yrs., Infectious Disease</div>

Providers noted that these staff shortages, in tandem with patient load, may disallow CHTC implementation. Another provider reported that their health facility is overwhelmed, such that this would stifle CHTC implementation due to the perceived inability to make any protocol changes and address the needs of the couple.

"…we're so full with what we have now, we don't have enough providers…I'm not against it….I'm not sure people have time, but if they had somebody who specializes in infectious disease, felt comfortable, and they had that person who was willing to do that, I think they [facility leadership] would welcome that. I just don't know that we have that right now."

ARNP, 10 yrs., Women's Health

***Community engagement and meeting patient needs:*** Provider endorsement of their health facility was also contingent on the existing patronage from the community, and the resources they currently can provide.

"….how can we get the word out to the community….just knowing that this is something that we are bringing to the forefront within the community…I think that will put us ahead of the game…we do have the HepC clinic, the HIV clinic and those are heavily advertised in the community…this would be you know just another great idea that it's being implemented…"

PA, 3 yrs., Generalist

Another aspect of facility preparedness for CHTC was not only the ability to screen a couple jointly, but actively refer patients to meet patients' needs, specifically in those settings that do not provide HIV care. Providers reported already having a facility-based protocol in place to refer those newly diagnosed to HIV care.

"If they do test positive for HIV, we always give results in person. We just make that a practice. And then we obviously want to follow the guidelines…not just sending the patient off to kind of figure out what's next…specifically calling an HIV care clinic that can offer that patient care right with the patient in the room, setting up the appointments, getting them engaged…"

ARNP, <10 yrs., Women's Health

## Discussion

Studies have determined that CHTC is an effective strategy to engage people not aware of their HIV status and those who are in HIV serodifferent relationships. Couple centered approaches to HIV, were also fundamental to demonstrating the efficacy of HIV biomedical interventions. Study findings here revealed that the current sample of health care providers overall endorsed CHTC as an HIV testing and care engagement modality. Other work had similar findings that both health consumers and health providers, support active couple-based modalities for HIV/STI prevention and care [11,30] as a way for mitigating transmission. Leblanc et al. (2019) found that 22 U.S. health providers endorsed CHTC and thought that it made sense when attempting to mitigate HIV transmission between partners and within communities. In the South Florida based sample, providers were a mix of both clinical and social service (HIV testers) and overall reported being willing to engage in CHTC implementation. As in the current sample, some had experience engaging couples/partners in STI screening and care and sexual health more broadly. In the South Florida sample, the clinical providers, who were mainly nurse practitioners, reported greater experiences with joint screening and care due to being in practices that focused on women in prenatal care or working in practices that serve people living with HIV. The South Florida setting is also a major urban center whereby the patient and provider population may be more diverse than the current sample.

The experiences and practices reported in the South Florida were motivated by provider's motivation to meet patient needs to facilitate STI disclosure and partner treatment [4,11]. Participants in the current study also reported a willingness to facilitate CHTC and were comfortable conceptualizing the strategy at their place of practice with some considerations. The salience of CHTC to providers practical experiences were in their descriptions of patients seeking HIV, but primarily STI services, whereby they felt compelled to engage patients and consider their patient's partners. In the current study, no provider reported engaging couples as routine practice for HIV or STI prevention and care. However most reported being adaptive in their practice to engage partners together either when requested by the patient to facilitate education or when there was perceived benefit to include the partner of patients in the diagnosis and treatment of STIs. Despite this practice no one in the current sample mentioned or spoke to existing guidelines like CHTC or even partner notification and expedited partner therapy to help mitigate STI transmission and address community-level persistence of HIV and other STIs.

Provider endorsement of CHTC seemed largely due to perceived enhancement of patient-provider engagement in HIV and other STI treatment and care, especially in the communication and dissemination of information among partners [31]. One of the most prominent features of provider CHTC endorsement and certainty is their belief in the strategy's ability to enhance sexual health literacy among patients and patient's partners. Providers reported that health literacy needs regarding HIV/STI testing and diagnosis, and STIs treatment regimens had bearing on the health of both partners. They noted that this warranted at times, a joint approach, and that CHTC may attend to provider burden by their not engaging in time-consuming, repetitive health education to mitigate HIV/STI transmission between partners. CHTC which encourages joint partner engagement in HIV/STI screening and care, can ensure that the STI and sexual health practices are patient-centered and can be tailored to the behaviors and specific health needs of patients and their partners. Providers with experience meeting couples/partners noted the benefit of receiving and giving health information and how this may enhance care. Research has shown that patient health literacy in HIV and other STI care has a bearing on the uptake of appropriate interventions and may be one of the most important elements contributing to persistent HIV/STI transmission [32]. Research has demonstrated that health literacy, in tandem with patient-provider communication and trust, has a bearing on the uptake of prevention and engagement in treatment strategies [31,33,34]. Therefore, provider practice that can enhance this and is built on trust between partners and with providers, warrant routinization in sexual health promotion practice. Studies have demonstrated the role of provider comfort in communication and certainty in biomedical and socio-behavioral strategies in promoting health literacy. One study, found that trust with providers was facilitated via provider communication which had bearing on the retention in care among Black women living with HIV infection [35]. However, adequate patient education can be perceived as time-consuming and challenged by competing priorities, and health systems that disallow robust patient engagement [36]. Given the importance of this feature in the context of STIs care, patient engagement modalities that can ease, expedite and assist in patient education and treatment, like CHTC, warrant consideration and uptake [37,38].

Provider uncertainty in a CHTC endorsement was based on provider's experiences with patient populations that have an exceptional susceptibility to HIV and other STIs. Historically provider uncertainty in the HIV prevention/STI care context was more to do with understanding and adopting HIV medication regimens for people living with HIV and provider competencies [39]. Current evidence of provider uncertainty is evident in suboptimal uptake and provision of HPV vaccination in the U.S. and globally. One systematic review found that health providers' perspectives on HPV and HPV vaccination impeded uptake, and that

personal biases may play a role [40]. Providers' uncertainty in the current study was related to the perceived inability to address potential couple discord in a joint health session and lack of transparency of partners' behaviors which impacts health education and appropriate care. Uncertainty was in part based on legitimate concerns of relationship violence and appeared to influence how some providers spoke about their facility's preparedness and hypothetical implementation of CHTC. Though research has shown that CHTC does not increase intimate partner violence in couples [41]. Provider uncertainty also involved whether communities would find CHTC an attractive strategy. Relationship discord and various dimensions of IPV between partners is a feature in STI transmission and recurrent infection [42]. However, research with couples have demonstrated even when this feature is present, couples/partners will engage in joint approaches to mitigate transmission [42]. Given this context, provider uncertainty in this regard would need to be addressed by ensuring that any provider training includes IPV screening and referral protocols for resources for couples to access. Research has demonstrated that when such guidelines are present within health facilities providers are motivated to engage in certain interventions resulting in improved patient outcomes. To address this and other causes for provider uncertainty, previous work on provider perception of CHTC, included a recommendation that a team-based approach is utilized to address issues like potential IPV and to leverage the skills and expertise of an interdisciplinary team in the clinical space [4]. Although no provider mentioned a team-based approach to implementation or current practices in their facility, such approaches may be worth curating in certain settings in Western-Central New York State.

Some established barriers to couples-based approaches include, power dynamics within relationships, and HIV stigma [43,44]. As one provider noted, although they fully endorsed the strategy and would promote it for their patients, due to HIV stigma, they questioned partners commitment in the context of a new HIV diagnosis. HIV stigma has been demonstrated to be one of the primary determinants of HIV prevention and care efforts [45]. However, research demonstrates that with provider support and adequate health literacy regarding antiretrovirals for prevention and treatment, partners who want to remain committed do [46]. It is also at these encounters whereby messaging regarding Undetectable=Untransmittable (U = U) and PrEP becomes crucial for patient and partners understanding and literacy about HIV/STI transmission, ART uptake, and maintenance of relationships should this be a goal [47]. However, this requires that providers are aware and comfortable with the concept of one, serodiscordance, then U = U and lastly the efficacy and diversity of PrEP relevancy for couples. Extant research demonstrates that there remains provider uncertainty in all these areas indicating research needs and intervention for this group of providers [39,47,48].

Lastly, provider uncertainty was related to provider perspectives of their facility's willingness and preparedness to implement CHTC, and whether institutional support and flexibility in existing protocols can allow for CHTC implementation. This flexibility and support pertained to the ability to see partners together and support staff facilitating some of the patient registration, scheduling processes, and testing. It was perceived that CHTC could eliminate multiple visits by a patient and their partner if they could be seen together. This could alleviate known issues with provider shortages and patient wait times for an appointment. Providers also noted billing of joint services being a concern and contemplated how services could be rendered with the current healthcare reimbursement structure. Seminal work conducted in South Florida revealed that providers believed CHTC to be a viable intervention, but also noted billing for CHTC and other partner services was also a concern and posed a barrier to CHTC implementation [11].

Akin to CHTC, other existing strategies like partner notification and expedited partner therapy (EPT) are opportunities to implement certain components of CHTC in that disclosure

and partner treatment are program goals Partner notification guidelines for certain STIs give direction to mitigating transmission and ensuring adequate treatment for partners of patients. EPT which like CHTC has demonstrated efficaciousness in reducing recurrent STIs among known partners, is a treatment specific modality for chlamydia and gonorrhea and no other STIs which may hold greater stigma, like herpes [7], Challenges to EPT utilization are multi-level and systematic, resulting in its underutilization, but that healthcare provider comfort and uncertainty plays an integral role in implementation. EPT can be optimized by being an entry for a CHTC implementation. In the current study, providers with couple/partner-based experiences to address STI diagnoses and treatment, and those without but who endorsed the approach demonstrated a level of comfort to engage patients and their partners jointly either in-person or remotely. They also demonstrated awareness of the importance of meeting patients' needs and CHTC may be a modality that does. The underutilization of existing national CHTC and EPT guidelines to address interpersonal attributes of HIV/STI transmission, in tandem with, provider uncertainty speaks to a need for enhanced provider training. Provider training would comprise that as offered within education curriculums and as continuing education opportunities, which would address provider uncertainty in management of STI prevention, specifically within couples. Provider uncertainty also spoke to health setting capacity to provide this type of service. So as healthcare providers enhance their training in CHTC implementation and related strategies, it is anticipated that health settings provision and standardization of CHTC could also be enhanced.

## Conclusion

This study contributes to efforts to end the HIV epidemic (EtHE), specifically provider endorsement and certainty for CHTC implementation. Health providers are central to the uptake of HIV/STI interventions hence, it is critical to understand their perspectives and perceived determinants for implementation. This study reported on the perspectives and practices of health providers in Western/Central New York State. This setting is unique because it consists of small urban centers experiencing continuous high burden of HIV/STI transmission, and hence may benefit from innovative approaches to mitigation that may be different from that implemented within larger urban settings, like New York City. Resources in settings such as these, may be more concentrated and serve populations who may reside in communities surrounding these small urban centers which may include semi-rural settings. This has implications to the persistent HIV/STI transmission, willingness to engage in certain preventative and treatment strategies over others. Couple-centered approaches like CHTC, expands health to be inclusive beyond the individual and has holistic interpersonal features which mitigates HIV/STI transmission, and further benefits in the uptake of biomedical treatment and prevention. Provider perspectives were cemented in their experience with engaging their couples susceptible to and seeking treatment for STIs including HIV. These perspectives are particularly important to improve HIV counseling and treatment, address HIV stigma and confront the rise in STIs. Findings from this current study indicate that providers have patient engagement experiences that involve patient's intimate partners in clinical practice. Such engagement or prospective engagement influenced their endorsement of CHTC as an approach that addresses stigma, health literacy, and interpersonal communication regarding HIV and other STIs. CHTC was perceived as a way of addressing not only HIV transmission but mainly other STIs, which have dramatically increased in Western New York, and enhancing patient-provider engagement to facilitate health literacy pertaining to STIs and sexual health more broadly.

Some limitations to note regarding this report is that interviews were initiated at the onset of COVID 2019 restrictions, and before the injectable formulation of PrEP was FDA approved.

However, the focus of this inquiry is regarding CHTC and findings should be assessed within this context. Many healthcare providers focused on addressing reduction in sexual health promotion in clinical settings and redirect their efforts to addressing the COVID pandemic. Most reports were based on past and recent practices; however, findings remain salient to current practices post-COVID restrictions in which HIV prevention and care remain heavily individualized and siloed. As efforts to end the HIV epidemic intensify, and jurisdictions across New York State experience substantial increases in bacterial and viral STIs, interventions to enhance HIV/STI screening and expedited treatment require greater consideration and implementation. CHTC is such a strategy that embodies U = U as a way to normalize HIV prevention and care, and has demonstrated efficacy in optimizing the HIV status-neutral continuum – a framework that centers HIV screening, and illuminates that regardless of one's HIV status, support is needed for people who are susceptible to HIV or who are in need of care [9,49]. This report sets a foundation for future investigation for CHTC and related strategies in U.S. health and community settings. Future health service research and efforts should address provider uncertainty with U = U and related concepts, demonstrate the feasibility and CHTC implementation experiences, and should address CHTC determinants.

## Supporting information

**S1 File. Interview guide.**
(DOCX)

## Author contributions

**Conceptualization:** Natalie M. Leblanc.

**Data curation:** Natalie M. Leblanc.

**Formal analysis:** Natalie M. Leblanc, Sadandaula R. Muheriwa-Matemba, Alexander Glazier, Faith Lambert, Martez Smith.

**Funding acquisition:** Natalie M. Leblanc.

**Investigation:** Natalie M. Leblanc.

**Methodology:** Natalie M. Leblanc.

**Supervision:** Natalie M. Leblanc.

**Writing – original draft:** Natalie M. Leblanc.

**Writing – review & editing:** Natalie M. Leblanc, Sadandaula R. Muheriwa-Matemba, Noelle St. Vil, Danielle Alcena-Stiner, Keosha T. Bond, Alexander Glazier, Luis Rosario-McCabe.

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
