## [Decision Letter · Decision Letter 0]

17 Jun 2024

PONE-D-24-02710Couple-Centered HIV Prevention and Care: Endorsement, Practice and Uncertainty among US Healthcare Providers in Western-Central Upstate New YorkPLOS ONE

Dear Dr. Leblanc,

Thank you for submitting your manuscript to PLOS ONE. After careful consideration, we feel that it has merit but does not fully meet PLOS ONE’s publication criteria as it currently stands. Therefore, we invite you to submit a revised version of the manuscript that addresses the points raised during the review process.

Please ensure that your manuscript follows the PLOS One journal's requirements for presentation and that all abbreviations are explained upon first use. Also, include a section discussing the study's limitations. And please, address the concerns raised by Reviewer 3, particularly regarding the consistency of terms, the potential impact of PrEP/PEP options on engagement, the currency of the interviews, and the lack of provider awareness of CHTC. Addressing these issues will strengthen your manuscript and enhance its impact.

We look forward to receiving your revised manuscript.

Kind regards,

Isaac Amankwaa, Ph.D.

Guest Editor

PLOS ONE

Journal Requirements:

3. Please expand the acronym “NIH/NIMH” (as indicated in your financial disclosure) so that it states the name of your funders in full.

4. We note that your Data Availability Statement is currently as follows: "All relevant data are within the manuscript and its Supporting Information files."

Reviewers' comments:

Reviewer's Responses to Questions

**Comments to the Author**

1. Is the manuscript technically sound, and do the data support the conclusions?

Reviewer #1: Partly

Reviewer #2: Yes

Reviewer #3: Yes

2. Has the statistical analysis been performed appropriately and rigorously? 

Reviewer #1: Yes

Reviewer #2: I Don't Know

Reviewer #3: N/A

3. Have the authors made all data underlying the findings in their manuscript fully available?

Reviewer #1: Yes

Reviewer #2: Yes

Reviewer #3: Yes

4. Is the manuscript presented in an intelligible fashion and written in standard English?

Reviewer #1: Yes

Reviewer #2: Yes

Reviewer #3: Yes

5. Review Comments to the Author

Reviewer #1: The authors have demonstrated good knowledge and understanding in the topic. However, the presentation did not follow the requirements of the Plos One journal. Also, the authors should consider and ensure a uniform font throughout. Again, there are some abbreviation that were used without explanation.

Reviewer #2: This is a well articulated manuscript. The qualitative methodology is well explained, and the results and discussion are sound. Please include a section on limitations of the study. Please resubmit after making the minor revision. Thank you.

Reviewer #3: Please see attached document with comments for authors. The manuscript is technically sound but could benefit from some additional minor additions and more importantly comparison with similar work done in a different geographic area in the US.

6. PLOS authors have the option to publish the peer review history of their article (what does this mean? ). If published, this will include your full peer review and any attached files.

**Do you want your identity to be public for this peer review?** For information about this choice, including consent withdrawal, please see our Privacy Policy .

Reviewer #1: No

Reviewer #2: No

Reviewer #3: No

---

## [Author Response · Author response to Decision Letter 0]

13 Dec 2024

Thank you reviewers for your insights. Please see attachments for response to reviewers. Thank you.

---

## [Editor Report · Decision Letter 1]

26 Dec 2024

Couple-Centered HIV Prevention and Care: Endorsement, Practice and Uncertainty among US Healthcare Providers in Western-Central Upstate New York

PONE-D-24-02710R1

Dear Dr. Natalie M. Leblanc,

We’re pleased to inform you that your manuscript has been judged scientifically suitable for publication and will be formally accepted for publication once it meets all outstanding technical requirements.

Kind regards,

Isaac Amankwaa, Ph.D.

Guest Editor

PLOS ONE

---

## [Editor Report · Acceptance letter]

PONE-D-24-02710R1

PLOS ONE

Dear Dr. Leblanc,

I'm pleased to inform you that your manuscript has been deemed suitable for publication in PLOS ONE. Congratulations! Your manuscript is now being handed over to our production team.

Kind regards,

on behalf of

Dr. Isaac Amankwaa

Guest Editor

PLOS ONE